# Differences in Frontal Lobe Dysfunction in Patients with Episodic and Chronic Migraine

**DOI:** 10.3390/jcm10132779

**Published:** 2021-06-24

**Authors:** Sang-Hwa Lee, Yeonkyeong Lee, Minji Song, Jae Jun Lee, Jong-Hee Sohn

**Affiliations:** 1Department of Neurology, Chuncheon Sacred Heart Hospital, Hallym University College of Medicine, Chuncheon 24523, Korea; neurolsh@hallym.or.kr (S.-H.L.); songminji@hallym.or.kr (M.S.); 2Institute of New Frontier Research, College of Medicine, Hallym University, Chuncheon 24523, Korea; leeyk10047@naver.com; 3Department of Psychology, College of Social Sciences, Hallym University, Chuncheon 24523, Korea; 4Department of Anesthesiology and Pain Medicine, College of Medicine, Kangwon National University, Chuncheon 24523, Korea; iloveu59@hallym.or.kr

**Keywords:** chronic migraine, episodic migraine, medication overuse headache, frontal lobe function

## Abstract

Neuroimaging and neuropsychological investigations have indicated that migraineurs exhibit frontal lobe-related cognitive impairment. We investigated whether orbitofrontal and dorsolateral functioning differed between individuals with episodic migraine (EM) and chronic migraine (CM), focusing on orbitofrontal dysfunction because it is implicated in migraine chronification and medication overuse headache (MOH) in migraineurs. This cross-sectional study recruited women with CM with/without MOH (CM + MOH, CM − MOH), EM, and control participants who were matched in terms of age and education. We conducted neuropsychological assessments of frontal lobe function via the Trail Making Test (TMT) A and B, the Wisconsin Card Sorting Test (WCST), and the Iowa Gambling Task (IGT). We enrolled 36 CM (19 CM + MOH, 17 CM − MOH), 30 EM, and 30 control participants. The CM patients performed significantly (*p* < 0.01) worse on the TMT A and B than the EM patients and the control participants. The WCST also revealed significant differences, with poorer performance in the CM patients versus the EM patients and the control participants. However, the net scores on the IGT did not significantly differ among the three groups. Our findings suggest that the CM patients exhibited frontal lobe dysfunction, and, particularly, dorsolateral dysfunction. However, we found no differences in frontal lobe function according to the presence or absence of MOH.

## 1. Introduction

Although cognitive symptoms are not considered among the core symptoms of migraines, many migraineurs often complain of cognitive dysfunction. Indeed, cognitive symptoms rank second after pain in terms of intensity and attack-related disability [1]. A clinical series on migraines reported that cognitive symptoms occurred in all phases of a migraine attack [2]. Clinical studies consistently report poor cognitive performance during migraine attacks, although the data regarding cognition in the interictal period are conflicting. Specifically, during the interictal period, most clinic-based studies have indicated that migraineurs show impaired cognitive function, whereas population-based studies have revealed no differences in cognitive function between migraineurs and controls [3]. Consistent with subjective patient complaints, executive function is most consistently affected during migraine attacks [3,4]. Functional neuroimaging studies of spontaneous or nitroglycerin-triggered migraine attacks have reported increased activity in cortical areas relevant to executive function, such as the cingulate cortex, insula, prefrontal cortex, and temporal poles [5]. Further, neuroimaging and neuropsychological investigations have identified frontal lobe-related brain abnormalities and cognitive impairment in migraineurs during the interictal period, although the relationship between brain anatomy and cognitive function is unclear [6].

Approximately 10% of individuals with episodic migraine (EM) develop chronic migraine (CM). CM is a disabling neurological condition characterized by an increase in the frequency and intensity of migraines. It often results in medication overuse headache (MOH), deterioration of patient quality of life, and represents a significant economic burden [7]. Previous studies have suggested that patients with CM exhibit orbitofrontal dysfunction, but whether this is influenced by medication overuse is still controversial [8,9]. Orbitofrontal dysfunction impairs reward prediction and decision-making mechanisms, which also occurs in substance abusers, and could contribute to MOH in CM patients [10,11]. Dorsolateral executive dysfunction has also been described in patients with EM and CM, although this is not clearly related to MOH [6,12]. Several studies have addressed cognitive abnormalities, particularly frontal-related cognitive dysfunction, in migraine patients between headache attacks. However, a consensus has not yet been established regarding frontal-related cognitive performance in these patients.

To address this in the present study, we investigated whether individuals with EM and CM differed in terms of frontal lobe-related cognitive function during the interictal period. We were particularly interested in orbitofrontal and dorsolateral function, as orbitofrontal dysfunction is related to migraine chronification and MOH in migraineurs.

## 2. Materials and Methods

### 2.1. Subjects

We recruited consecutive female patients who had CM with/without MOH (CM + MOH, CM − OH), EM, and control participants from the neurology outpatient department of a university hospital. All participants were females between 20 and 60 years of age to eliminate age and sex bias. A board-certified neurologist used patient history, neurological examinations, laboratory or neuroimaging data, and the International Classification of Headache Disorders-3 (ICHD-3) criteria to classify headache diagnoses. To exclude other primary sources of headache, the patients were required to have at least a one-year history of migraine headaches before enrollment. The control group consisted of age- and education-matched volunteers. We recruited the control group by inviting people who accompanied the patients to join the study (e.g., friends), and also posted advertisements (e.g., posted notices in the hospital). The controls had been free of headaches for at least three months before the study, experienced no more than an occasional mild headache (<5 times per year), and had not sought medical treatment for headaches. The exclusion criteria were the presence of neurological disorders (e.g., stroke), pregnancy, serious somatic or psychiatric illness including depression and anxiety disorder, daily medication to prevent headaches, and/or psychoactive medication regimes such as antidepressants within the last three months.

All participants underwent physical and neurological examinations, which were performed by an experienced neurologist. The participants were asked to complete a questionnaire regarding their headache symptoms, including frequency, duration, and intensity, during the previous three months. Headache frequency (days/month) was calculated by dividing the number of days with headaches by three months. Headache duration (hours/day) was calculated by dividing the sum of the total hours of headache by the number of days with a headache. Headache intensity (numeric rating scale [NRS]: 0 = no pain to 10 = unbearable pain) was calculated as the mean NRS score for the days with a headache. In addition, we recorded the number of days per month, the number of times medications were used, as well as the types of medications used including analgesics, nonsteroidal anti-inflammatory drugs (NSAIDs), ergot, triptans, and opioids. All participants gave their written informed consent before enrollment. The university hospital ethics committee approved this study.

### 2.2. Neuropsychological Evaluation

All migraineurs underwent a clinical interview carried out by a team of neurologists and psychiatrists. To accurately assess the impact of headaches on the lives of the patients, we asked them to complete the Headache Impact Test-6 (HIT-6) and the Migraine Disability Assessment Scale (MIDAS). In addition, each patient completed a self-administered Patient Health Questionnaire-9 (PHQ-9) to assess depression and the Generalized Anxiety Disorder-7 (GAD-7) to assess anxiety.

To assess differences in frontal lobe-related cognitive function during the interictal period, we used neuropsychological test data collected at least three days after the last migraine and three days before the next migraine attack in the EM groups. All of the CM patients also completed the neuropsychological test in the interictal period (at least three days before and three days after a typical migraine attack), but we still used their data if they presented a current background mild headache (NRS < 3). To ascertain the length of the interictal period, patients were interviewed by telephone three days after the neuropsychological assessment. In addition, headache intensity was assessed via NRS during the examination. The subjects were instructed to report for examination on the days in which their headache intensity was <3 points on an NRS, and this was especially for the CM patients. The assessment of frontal lobe function included the Trail Making Test (TMT) A and B, Wisconsin Card Sorting Test (WCST), and Iowa Gambling Task (IGT).

The TMT provides information regarding visual search, scanning, processing speed, mental flexibility, and executive function [13]. We used the Korean version of the TMT (K-TMT) [14,15]. The test includes type A and type B tasks, and the time taken to complete the assignment is measured. In the type A task, numbers are arranged irregularly on a page, and the participant is asked to connect them in the correct order using lines. In type B, the participant is asked to connect numbers and letters alternately in order. In both tasks, the participant is asked to complete the task as quickly as possible. This test evaluates frontal lobe function because it requires intact visual perception, visual scanning, continuous attention, psychomotor speed, and attention shifting ability. Type B of the TMT is particularly appropriate for evaluating the executive function that depends on the dorsolateral frontal region.

The WCST evaluates the ability to think flexibly, to form abstract concepts, to make tactical plans, to respond to feedback, to modify plans to achieve goals, and to control impulse responses. The test consists of four different stimulus cards and sixty-four response cards. The participant is given response cards one at a time. They are asked to find a common feature between the response card and one of the four stimulus cards placed in front of them and to place the response card under the appropriate stimulus card. The common features are color, shape, and number. When the subject places the response card under a stimulus card, the examiner does not provide feedback about the correct answer but tells the participant whether their response is correct or incorrect. Whether the subject can respond to the stimulus while flexibly modifying their plan according to the examiner’s feedback reflects activity in the dorsolateral area of the frontal lobe. In this study, we counted the total number of correct responses (TC), the total number of error responses (TE), the number of persistent responses (PR), the number of persistent errors (PE), the number of conceptual level responses (CL), the number of categories completed (CC), and the number of trials to complete the first category (TCFC) [16,17]. A higher number of TC reflects stronger abstract generalization, working memory, attention, and executive control ability [17]. Greater TE and PE reflect decreased patient cognitive transfer and executive control function, as well as reduced cognitive flexibility. A greater number of completed classifications imply enhanced concept conversion, classification initiative, and comprehension of diversity [18].

In addition, to test decision-making mediated by orbitofrontal function, we administered the computer-based IGT to the CM patients, the EM patients, and the healthy control participants. This task is very sensitive to ventromedial and orbitofrontal lesions of the prefrontal cortex [19]. It consists of picking cards from four decks, each of which can result in an unpredictable reward or penalty. In this task, the four decks are presented on a computer monitor. When a deck is selected, numbers representing gains and losses are presented on the upper part of the monitor. Gains and losses occur in certain ratios. Decks A and B deliver large immediate gains but more losses than gains in the long run, whereas decks C and D deliver small immediate gains but more gains than losses overall. The sequence of gains and losses and the amount of each gain or loss encountered with each of the four decks are fixed [20]. Participants are given no instruction regarding which decks are advantageous or disadvantageous. Instead, they are instructed to gain as many points as possible before the completion of the task. The net score is represented by the number of times that advantageous choices (C + D) are made minus the number of disadvantageous choices (A + B). In other words, if the value of (C + D) − (A + B) is positive, then the examinee has chosen more advantageous than disadvantageous cards, and if negative, the opposite is true. The IGT net scores range from −100 to +100. A negative net score indicates a decision-making deficit [21]. We analyzed each of the variables of the TMT (TMT A, TMT B) and WCST (TC, TE, PR, PR, CL, CC, and TCFC), as well as the net score of the IGT, and compared these among the three groups.

### 2.3. Statistical Analysis

All values are reported as means and standard deviations. Demographics, headache characteristics, and affective features were compared between the groups using a one-way analysis of variance (ANOVA) or the Student’s *t*-test. We used an ANOVA with a Bonferroni test as post hoc analyses to compare neuropsychological performance among the EM, CM, and control groups. In addition, Pearson correlation analysis was used to explore the relationship between the neuropsychological and the clinical parameters. To examine differences in the demographics and neuropsychological data in the CM patients with and without MOH, we performed the Student’s *t*-test. A *p*-value < 0.05 was considered statistically significant. SPSS software (SPSS for Windows, ver. 23.0; SPSS, Chicago, IL, USA) was used for all comparisons.

## 3. Results

### 3.1. Clinical Characteristics

In total, 36 patients with CM (19 CM + MOH, 17 CM − MOH), 30 patients with EM, and 30 control participants were enrolled in the study. The demographic, clinical, and affective characteristics of the patient population are described as means and standard deviations and reported in Table 1. There were no significant differences in age or years of education among the three groups. The CM patients had significantly higher levels of depression, as measured using the PHQ-9, and higher levels of headache-related disability, as measured using the MIDAS, compared to the EM patients. There were no significant differences in HIT-6 and GAD-7 scores between the patients with EM and CM. Of the 19 enrolled patients with CM + MOH, the types of medication used were as follows; simple analgesics in seven patients, NSAIDs in five, triptans in two who also used NSAIDs, and combination analgesics in seven.

### 3.2. Neuropsychological Tests

To assess neuropsychological differences in frontal lobe function, we administered the TMT A and B. We found that the CM patients performed significantly poorer than the EM patients and the control participants in both the TMT A and B (*p* < 0.01). In terms of dorsolateral function, variables measured using the WCST (TC, TE, CL, and CC) revealed significant differences, with poorer performance in the CM patients compared to the EM patients and the control participants (*p* < 0.01). Although other parameters of the WCST (PR, PE, and TCFC) were significantly different among the three groups (*p* < 0.05), post hoc analysis revealed no significant differences within each group. In terms of orbitofrontal function, which was tested via the IGT, performance in the CM group was poorer than that in the EM and the control groups. We found no significant differences in the net IGT scores among the three groups, although the CM group exhibited poorer performance. The neuropsychology test results for the migraineurs (episodic, chronic) and controls are displayed in Table 2.

We also compared the demographic and neuropsychological data between the CM patients with and without MOH. We found no differences in each neuropsychological test component according to the presence or absence of MOH. In terms of the mean values of IGT net scores, patients with CM + MOH had negative net scores and patients with CM − MOH had positive net scores. However, these differences were not significant (Table 3). Further, we found no significant differences between the EM and the CM + MOH groups in terms of decision-making processes mediated by the orbitofrontal cortex, as measured using the IGT.

### 3.3. Correlation between Headache Parameters and Neuropsychological Variables

We analyzed the correlation between the headache parameters such as the monthly number of days with a headache, duration, intensity, days per month with medication intake, and each neuropsychological variable. We found significant correlations between the monthly days with a headache and the TMT variables (TMT A 0.507, *p* = 0.000, TMT B 0.323, *p* = 0.013). In addition, we found a significant relationship between the number of the monthly days with a headache and the components of the WCST (TC 0.290, *p* = 0.026, TE 0.279, *p* = 0.032, PR 0.275, *p* = 0.035, PE 0.280, *p* = 0.032, CL 0.316, *p* = 0.015), although this was not the case for the IGT. Further, other headache parameters (duration, intensity, days per month, and number of times medication was taken) were not significantly correlated with frontal lobe-related neuropsychological variables (TMT, WCST, IGT). We used correlation analysis to explore the possible relationships between frontal lobe-related neuropsychological variables and affective scores. We only found significant correlations between TMT components and depression scores measured using the PHQ-9 (TMT A 0.347, *p* = 0.007, TMT B 0.329, *p* = 0.011). Pearson’s correlation coefficients revealed no significant relationships between the net IGT scores and affective characteristics across the EM and the CM groups.

## 4. Discussion

We prospectively recruited female CM patients with/without MOH, EM patients, and age- and education-matched control participants, and administered neuropsychological assessments of frontal lobe function using the TMT, WCST, and IGT during the interictal period. We found that, compared to the EM patients and the control participants, performance in the CM patients was significantly poorer in the TMT A and B, as well as several WCST components. However, the net IGT scores were not significantly different between the groups. Thus, the CM group exhibited poorer frontal-related cognitive performance, particularly in terms of dorsolateral executive function, compared to the EM and the control groups. However, we found no significant differences in orbitofrontal function between the CM + MOH and the CM − MOH groups or between the CM + MOH and the EM groups.

Our results are consistent with previous findings. One study reported that all types of migraine headaches, including those without an aura, are associated with poorer cognitive performance compared to an absence of a headache. Particularly, significantly poorer performance on the TMT B in migraineurs suggests impaired executive function, attention, and processing speed [22]. Other studies have shown that migraineurs perform worse on a sustained attention task and exhibit low processing speed, as measured using the TMT. However, performance on a verbal fluency task, working memory as measured using the Stroop test, and measures of verbal and visual learning and recall are not affected [23]. The majority of cross-sectional clinic-based studies that have performed cognitive testing during the interictal period in patients with EM have reported decreased performance in migraineurs, either with or without auras, compared to controls in several cognitive domains, such as attention, executive function, language, memory, motion perception, processing speed, and visuospatial memory [6,12,24,25,26,27,28,29,30,31]. Disease severity parameters such as frequency, duration, intensity of headache attacks, and overall disease duration may influence cognitive impairment in migraineurs. Several studies have demonstrated that cognitive performance is independent of these clinical parameters [29,32], whereas others have reported that cognitive ability depends on the frequency [26,30] or intensity [28,30] of headache attacks.

Unlike previous studies, we compared differences in frontal lobe-related cognitive function between the EM and the CM patients during the interictal period and analyzed correlations between the headache parameters and the various neuropsychological variables. We found significant correlations between headache frequency, as measured using the number of headache days per month, and components of neuropsychological tests such as the TMT and the WCST. Similarly, two previous studies indicated that cognitive performance in migraineurs is related to headache frequency [27,31]. One study that used an extensive cognitive battery showed that all test scores declined with increasing headache frequency, while attention, memory, and visuomotor speed processing were particularly affected in high-frequency migraineurs [27]. Another study measured cognitive performance via the Montreal Cognitive Assessment and event-related potentials and reported that migraineurs performed worse on tests of language, verbal and visual memory, executive function, calculation, and orientation, and that executive dysfunction was particularly related to headache frequency [31]. However, these previous studies did not exclude patients with psychiatric disorders and those who were taking psychoactive medications. Previous studies reported that clinical factors influence cognitive function, including psychological problems (depression, chronic stress, exhaustion, and sleeping problems) [33], sex (especially during pregnancy or menopause) [34,35], preventive medication [36], and age [37]. Thus, we excluded patients with serious somatic or psychiatric illnesses, including depression and anxiety disorder, as well as those taking daily medication to prevent headaches and/or psychoactive drugs such as antidepressant medication in the present study. Despite the exclusion criteria, the severity of depressive symptoms, as measured using the PHQ-9, affected attention and processing speed, as measured using the TMT A and B. In recent studies, subjective cognitive complaint scores tended to increase with the frequency of migraines with aura, and this interrelation is influenced by depression severity [38]. In a five-year longitudinal study, migraine was not associated with an increased risk of dementia or cognitive decline in older age, but individuals with migraines had more subjective cognitive complaints and depressive symptoms than the control patients did [39].

We administered the IGT to investigate orbitofrontal dysfunction, which is related to migraine chronification and MOH in migraineurs. However, we found no significant differences in orbitofrontal function between the CM + MOH and the CM − MOH groups or between the CM + MOH and the EM groups, although the mean IGT scores in the CM + MOH patients were negative, while those in the CM − MOH and the EM patients were positive. Previous studies have indicated that orbitofrontal dysfunction, as revealed by neuropsychological assessment, is present in the CM patients with MOH [8,9,40,41]. In terms of follow-up evaluations, some studies have reported that negative outcomes are associated with poor baseline orbitofrontal performance [40,41], whereas others have suggested that persistent orbitofrontal dysfunction is not influenced by the presence of MOH [8,9]. However, several imaging studies have reported localized anatomical and functional brain changes in patients with MOH. Some data suggest that MOH is associated with functional changes within intrinsic brain networks rather than with macrostructural change [42,43]. Others have found that patients with CM + MOH versus CM − MOH show a decrease in gray matter volume in the orbitofrontal cortex and the middle occipital gyrus or the middle temporal gyrus, which are involved in avoidant and addictive behaviors, respectively [44,45]. Particularly, the gray matter volume of the orbitofrontal cortex is predictive of the response to MOH treatments [45]. In addition, several studies have investigated brain structure and function before and following withdrawal of migraine medication overuse. These have found that for brain regions that are related to pain processing, those that appear abnormal during medication overuse return to normal following discontinuation of the overused medications. However, for regions that are implicated in the pathophysiology of addiction, such abnormalities tend to persist even after the discontinuation of the overused medications. This suggests that abnormal function in these regions might predispose an individual to medication overuse, or, conversely, that the overuse of medications can result in long-standing abnormalities within these brain regions [42,46,47,48]. Recent studies found that CM − MOH during the interictal phase is associated with functional connectivity alterations in regions involved in multisensory integration, affective and cognitive processing, and pain modulation [49]. Another study of the functional characteristics of the brain in CM − MOH during the interictal phase, using static functional connectivity and static and dynamic functional network connectivity analyses, found that the abnormal connectivity pattern between sensory and cognitive brain networks and altered connectivity were concentrated in the executive control network of the CM patients [50]. Compared to patients with a migraine, patients with MOH exhibit exacerbated changes in brain structure and function in regions of the pain matrix and areas of the mesocortical-limbic circuit [51].

In our study, the net IGT scores were somewhat higher compared to previous studies [8,9]. However, IGT performance was not related to the headache parameters or the affective factors such as GAD-7 or PHQ-9 scores, similar to the results of a previous study [8]. Therefore, further neuropsychological studies with follow-up assessments are needed to determine the role of frontal lobe dysfunction in migraine chronification and medication overuse.

There are several limitations to our study that should be considered. First, our sample of female migraineurs and control subjects was relatively small. Second, the study sample was recruited in a specialized headache clinic at one university hospital, which might have led to a selection bias. Thus, these findings may not be truly representative of migraine patients and normal controls, or generalizable to other groups. Moreover, due to the cross-sectional nature of our study, we were unable to examine casual relationships. Further longitudinal follow-up studies are needed to assess causality.

## 5. Conclusions

We found evidence of frontal lobe dysfunction, particularly dorsolateral dysfunction, in the CM patients compared to the EM patients and the control participants during the interictal period. However, we found no differences in frontal lobe function according to the presence or absence of MOH in the CM patients. Few studies have examined cognitive performance, particularly frontal lobe-related cognitive impairment, in CM patients. Thus, clinicians should consider frontal lobe dysfunction in migraine patients, even during headache-free periods. In addition, comprehensive studies using functional neuroimaging and neuropsychological tests, along with clinical observation, are necessary to elucidate frontal lobe dysfunction in migraineurs.

## Figures and Tables

**Table 1 jcm-10-02779-t001:** Demographic, headache, and affective characteristics of the participants.

	Episodic Migraine (*n* = 30)	Chronic Migraine (*n* = 36)	Control (*n* = 30)	*p*-value
Age (years)	41.57 ± 8.68	47.86 ± 11.49	43.03 ± 13.71	ns
Education (years)	12.65 ± 3.28	10.93 ± 3.06	12.23 ± 2.55	ns
Headache frequency (days/month)	5.09 ± 3.56	23.53 ± 6.26		0.000
Headache duration (hours/day)	23.37 ± 23.69	22.58 ± 22.27		ns
Headache intensity (NRS)	7.91 ± 1.56	7.28 ± 2.03		ns
Days of drug intake (days/month)	3.70 ± 3.69	14.33 ± 11.40		0.000
MIDAS total scores	20.61 ± 23.13	52.44 ± 51.49		0.007
HIT-6 total scores	59.30 ± 7.50	60.94 ± 13.20		ns
GAD-7	7.87 ± 5.39	10.33 ± 6.17		ns
PHQ-9	8.30 ± 6.48	13.06 ± 6.01		0.006

Values expressed as mean ± SD; numeric rating scale, NRS; Migraine Disability Assessment Scale, MIDAS; Headache Impact Test-6, HIT-6; Generalized Anxiety Disorder-7, GAD-7; Patient Health Questionnaire-9, PHQ-9; nonsignificant, ns.

**Table 2 jcm-10-02779-t002:** Neuropsychology test results for migraineurs (episodic, chronic) and the controls.

	Control ^a^ (*n* = 30)	Episodic Migraine ^b^ (*n* = 30)	Chronic Migraine ^c^ (*n* = 36)	*p*-Value	*Post-hoc*
TMT: A	31.22 ± 23.76	28.00 ± 7.78	48.56 ± 25.95	0.001	a = b < c
TMT: B	62.30 ± 24.71	76.17 ± 37.06	109.50 ± 59.62	0.001	a = b < c
WCST: TC	50.70 ± 5.36	50.09 ± 4.04	44.58 ± 8.30	0.000	a = b > c
WCST: TE	13.30 ± 5.36	13.91 ± 4.04	19.31 ± 8.29	0.000	a = b < c
WCST: PR	7.23 ± 3.81	7.22 ± 2.78	9.44 ± 4.62	0.038	ns
WCST: PE	6.97 ± 3.31	6.87 ± 2.53	8.75 ± 3.87	0.049	ns
WCST: CL	47.93 ± 7.20	47.22 ± 6.30	39.58 ± 10.81	0.000	a = b > c
WCST: CC	3.97 ± 0.89	3.78 ± 0.85	3.03 ± 1.23	0.001	a = b > c
WCST: TCFC	13.03 ± 3.63	13.35 ± 5.51	16.11 ± 6.09	0.037	ns
IGT: net score	14.80 ± 28.54	6.87 ± 21.39	3.06 ± 20.39	ns	ns

Values expressed as mean ± SD; ^a^ values of control group; ^b^ values of Episodic Migraine group; ^c^ values of Chronic Migraine group; Trail Making Test, TMT; Wisconsin Card Sorting Test, WCST; total number of correct responses, TC; total number of error responses, TE; number of persistent responses, PR; number of persistent errors, PE; number of conceptual level responses, CL; number of categories completed, CC; number of trials to complete the first category, TCFC; Iowa Gambling Task, IGT; nonsignificant, ns.

**Table 3 jcm-10-02779-t003:** Demographics and neuropsychological test results of chronic migraine patients with and without medication overuse headaches.

	CM − MOH (*n* = 17)	CM + MOH (*n* = 19)	*p*-Value
Age (years)	48.76 ± 10.44	47.05 ± 12.59	ns
Education (years)	10.73 ± 3.58	11.05 ± 2.61	ns
TMT: A	48.82 ± 20.09	48.32 ± 30.83	ns
TMT: B	122.35 ± 67.40	98.00 ± 50.79	ns
WCST: TC	44.59 ± 8.62	44.58 ± 8.24	ns
WCST: TE	19.41 ± 8.62	19.21 ± 8.22	ns
WCST: PR	9.59 ± 4.54	9.32 ± 4.81	ns
WCST: PE	8.94 ± 3.72	8.58 ± 4.10	ns
WCST: CL	40.06 ± 11.05	39.16 ± 10.87	ns
WCST: CC	2.94 ± 1.30	3.11 ± 1.20	ns
WCST: TCFC	15.24 ± 5.61	16.89 ± 6.54	ns
IGT: net scores	7.88 ± 20.55	−1.26 ± 19.79	ns

Values expressed as mean ± SD; chronic migraine without medication overuse headache, CM − MOH; chronic migraine with medication overuse headache, CM + MOH; Trail Making Test, TMT; Wisconsin Card Sorting Test, WCST; total number of correct responses, TC; total number of error responses, TE; number of persistent responses, PR; number of persistent errors, PE; number of conceptual level responses, CL; number of categories completed, CC; number of trials to complete the first category, TCFC; Iowa Gambling Task, IGT; nonsignificant, ns.

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
