# Peer review of "Differences in Frontal Lobe Dysfunction in Patients with Episodic and Chronic Migraine"

_jcm, 2021, doi:10.3390/jcm10132779_

Round 1

Reviewer 1 Report

The manuscript “Differences in Frontal Lobe Dysfunction in Patients with Episodic and Chronic Migraine” is a well-written substantive paper on disorders of cognitive functions in migraine patients, especially with chronic one. Neuropsychological methods and tests are well chosen, supported by adequate statistical methods and interesting critical conclusions. It is true that the study group is not large, but the study can be treated as preliminary, as the authors write in their “limitations part”. The results are very interesting and  in the general trend confirming that migraine is a disease that significantly impairs everyday functioning, also in the interictal (headache-free) periods.

Nevertheless, the authors should modify the literature according to the most recent -the most recent citation and only one comes from 2019, but in the last two years there have been papers on fMRI, the neural network and the debilitating effects of migraine on life, although little on neuropsychological impact in migraine. The manuscript requires minor corrections regarding English language.

Author Response

Dear Reviewer 1,

Please find attached a revised version of our manuscript, “Differences in Frontal Lobe Dysfunction in Patients with Episodic and Chronic Migraine” (jcm-1251291).

We thank you for your thoughtful suggestions regarding the original version of our paper; most of the suggested changes have been incorporated into the revision.

All revisions are described in detail in the order mentioned in the review, following the reviewer’s critique in italics. We believe that the revisions have greatly improved the manuscript and hereby submit the revised version for your consideration for publication.

Comments to author:

The manuscript “Differences in Frontal Lobe Dysfunction in Patients with Episodic and Chronic Migraine” is a well-written substantive paper on disorders of cognitive functions in migraine patients, especially with chronic one. Neuropsychological methods and tests are well chosen, supported by adequate statistical methods and interesting critical conclusions. It is true that the study group is not large, but the study can be treated as preliminary, as the authors write in their “limitations part”. The results are very interesting and in the general trend confirming that migraine is a disease that significantly impairs everyday functioning, also in the interictal (headache-free) periods.

We thank the reviewer for these comments and specific suggestions, which have improved our manuscript.

Nevertheless, the authors should modify the literature according to the most recent -the most recent citation and only one comes from 2019, but in the last two years there have been papers on fMRI, the neural network and the debilitating effects of migraine on life, although little on neuropsychological impact in migraine.

As recommended, we have added several sentences and revised the Discussion as follows:

In recent studies, subjective cognitive complaint scores tended to increase with the frequency of migraines with aura, and this interrelation is influenced by depression severity [38]. In a 5-year longitudinal study, migraine was not associated with an increasing risk of dementia or cognitive decline at an older age, but individuals with migraine had more subjective cognitive complaints and depressive symptoms than did controls [39].

(page 7, lines 304 – page 7, lines 309)

Recent studies found that CM-MOH during the interictal phase is associated with functional connectivity alterations in regions involved in multisensory integration, affective and cognitive processing, and pain modulation [49]. Another study of the functional characteristics of the brain in CM-MOH during the interictal phase using static functional connectivity and static and dynamic functional network connectivity analyses found that the abnormal connectivity pattern between sensory and cognitive brain networks and altered connectivity were concentrated in the executive control network of CM patients [50]. Compared to patients with migraine, patients with MOH exhibit exacerbated changes in brain structure and function in regions of the pain matrix and in areas of the mesocortical-limbic circuit [51].

(page 8, lines 346 – page 8, lines 355)

We have also added this citation:

  1. Chu, HT.; Liang, CS.; Lee, JT.; Lee, MS.; Sung, YF.; Tsai, CL.; Tsai, CK.; Lin, YK.; Ho, TH.; Yang, FC. Subjective cognitive complaints and migraine characteristics: A cross-sectional study. Acta Neurol Scand 2020,141,319–327, DOI: 10.1111/ane.13204.
  2. Martins, IP.; Maruta, C.; Alves, PN.; Loureiro, C.; Morgado, J.; Tavares, J.; Gil-Gouveia, R. Cognitive aging in migraine sufferers is associated with more subjective complaints but similar age-related decline: a 5-year longitudinal study. JHP 2020,21,31, https://doi.org/10.1186/s10194-020-01100-x.
  3. Dai, L.; Yu, Y.; Zhao, H.; Zhang, X.; Su, Y.; Wang, X.; Hu, S.; Dai, H.; Hu, C.; Ke, J. Altered local and distant functional connectivity density in chronic migraine: a resting-state functional MRI study. Neuroradiology 2021, 63, 555–562, https://doi.org/10.1007/s00234-020-02582-x.
  4. Zou, Y.; Tang, W.; Qiao, X.; Li, J.; Aberrant modulations of static functional connectivity and dynamic functional network connectivity in chronic migraine. Quant Imaging Med Surg 2021,11(6),2253-2264, http://dx.doi.org/10.21037/qims-20-588.
  5. Chong CD.; Brain structural and functional imaging findings in medication-overuse headache. Front Neuro 2020,10,1336, doi: 10.3389/fneur.2019.01336.

The manuscript requires minor corrections regarding English language.

The paper has been completely checked by English-language specialists of www.textcheck.com. Textcheck is an internet-based editing service offering English-language support to academics worldwide. The English in this document has been checked by at least two professional editors, both native speakers of English. For a certificate, please see: http://www.textcheck.com/certificate/TOUyTC

http://www.textcheck.com/certificate/JDNH4R

We have tried to address the issues raised by the reviewers and editorial board member. We are grateful for the constructive comments that arose during the review process. We believe that our paper has been improved based on these suggestions.

Yours faithfully,

Reviewer 2 Report

The authors aim at evaluating the functioning of frontal lobe in patients with migraine, under the hypothesis that there is a malfunctioning of the dorsolateral and orbitofrontal cortices in patients with episodic and chronic migraine; there is literature in that direction.

They have studied 3 groups of subjects: episodic migraine, chronic migraine (with and without medication overuse), and controls.  They have administered a task to evaluate the dorsolateral cortex function, the TMT. The WCST, also administered to evaluate the DL function, is also implicated in orbitofrontal function (perseverative errors). Finally, they used the IGT to assess the OF function. They found differences in DL functioning in patients with CM (either with or without medication overuse) as compared with controls and episodic migraine patients.

Therefore, while trying the hypothesis that OF is malfunctioning in chronic migraine, they find a DL malfunctioning. These findings correlate with some of the variables (migraine days), but not with others (use of medication).

The design of this study seems more a cross-sectional than a prospective study, and this has to be clarified and perhaps modified.

It was surprising to me that none of the patients were on preventive mediation. Is this correct?

The authors should clarify the medication used by the patients. How could the use of opioids have influenced their results (frontal lobe tests)? Could this explain, at least in part, that the worse group was that of chronic migraine patients (i. e, they consume more medication)?.

The test employed to assess the OF function, the IGT, does not correlate with any clinical or affective parameter, there are no differences between groups, and the standard deviations are important. Could this represent that this is a difficult to perform test and was performed at random by most of the participants?

Several paragraphs of the discussion do not add great information to this paper and can be dispensed with (lines 293-302).

Author Response

Dear Reviewer 2,

Please find attached a revised version of our manuscript, “Differences in Frontal Lobe Dysfunction in Patients with Episodic and Chronic Migraine” (jcm-1251291).

We thank you for your thoughtful suggestions regarding the original version of our paper; most of the suggested changes have been incorporated into the revision.

All revisions are described in detail in the order mentioned in the review, following the reviewer’s critique in italics. We believe that the revisions have greatly improved the manuscript and hereby submit the revised version for your consideration for publication.

Comments to author:

The authors aim at evaluating the functioning of frontal lobe in patients with migraine, under the hypothesis that there is a malfunctioning of the dorsolateral and orbitofrontal cortices in patients with episodic and chronic migraine; there is literature in that direction.

They have studied 3 groups of subjects: episodic migraine, chronic migraine (with and without medication overuse), and controls. They have administered a task to evaluate the dorsolateral cortex function, the TMT. The WCST, also administered to evaluate the DL function, is also implicated in orbitofrontal function (perseverative errors). Finally, they used the IGT to assess the OF function. They found differences in DL functioning in patients with CM (either with or without medication overuse) as compared with controls and episodic migraine patients.

Therefore, while trying the hypothesis that OF is malfunctioning in chronic migraine, they find a DL malfunctioning. These findings correlate with some of the variables (migraine days), but not with others (use of medication).

We thank the reviewer for these comments and specific suggestions, which have improved our manuscript.

The design of this study seems more a cross-sectional than a prospective study, and this has to be clarified and perhaps modified.

We agree with this important comment. Ours is a cross-sectional study rather than a prospective cohort study. Thus, we now describe the limitations of a cross-sectional design in the Discussion as follows: “Moreover, due to the cross-sectional nature of our study, we were unable to examine casual relationships. Further longitudinal follow-up studies are needed to assess causality.” For clarity, we have revised the following sentences in the Abstract and Materials and Methods as follows:

This cross-sectional study recruited women with CM with/without MOH (CM+MOH, CM-MOH), EM, and controls who were matched in terms of age and education.

(page 1, lines 19 – page 1, lines 21)

We recruited consecutive female patients who had CM with/without MOH (CM+MOH, CM-MOH), EM, and controls from the neurology outpatient department of a university hospital.

(page 2, lines 71 – page 2, lines 73)

It was surprising to me that none of the patients were on preventive mediation. Is this correct?

Migraine prophylactic medications (e.g., topiramate) or prevalent comorbidity (e.g., depression and anxiety) can also contribute to cognitive impairment in migraineurs. Previous studies have reported that clinical variables may influence cognitive function, including psychological problems (depression, chronic stress, exhaustion, and sleeping problems), sex (especially during pregnancy or menopause), preventive medication, and age. To exclude these effects and differentiate our study from previous studies, we recruited only new patients who visited the hospital for the study; among them, patients who took migraine prophylaxes or psychiatric medications within 3 months were excluded. Accordingly, we have revised and added the following text to the Materials and Methods and Discussion as follows:

The exclusion criteria were the presence of neurological disorders (e.g., stroke), pregnancy, serious somatic or psychiatric illness including depression and anxiety disorder, daily medication to prevent headaches, and/or psychoactive medication regimes such as antidepressants within the last 3 months. (page 2, lines 84 – page 2, lines 87)

Previous studies reported that clinical factors influence cognitive function, including psychological problems (depression, chronic stress, exhaustion, and sleeping problems) [33], sex (especially during pregnancy or menopause) [34, 35], preventive medication [36], and age [37].

(page 7, lines 296 – page 7, lines 299)

Also, we have added this citation:

  1. Stenfors, CU.; Marklund, P.; Hanson, LLM.; Theorell, T.; Nilsson, L-G. Subjective cognitive complaints and the role of executive cognitive functioning in the working population: a case-control study. PLoS ONE 2013, 8(12), e83351, doi: 10.1371/journal.pone.0083351. eCollection 2013.
  2. Onyper, SV.; Searleman, A.; Thacher, PV.; Maine, EE.; Johnson, AG. Executive functioning and general cognitive ability in pregnant women and matched controls. J Clin Exp Neuropsychol 2010,32(9), 986-995, doi: 10.1080/13803391003662694.
  3. Drogos LL, Rubin LH, Geller SE, Banuvar S, Shulman LP, Maki PM. Objective cognitive performance is related to subjective memory complaints in midlife women with moderate to severe vasomotor symptoms. Menopause 2013,20(12),1236-1242, doi: 10.1097/GME.0b013e318291f5a6.
  4. Kececi H, Atakay S. Effects of topiramate on neurophysiological and neuropsychological tests in migraine patients. J Clin Neurosci 2009,16(12),1588-1591, doi: 10.1016/j.jocn.2009.03.025.
  5. Samson RD, Barnes CA. Impact of aging brain circuits on cognition. Eur J Neurosci 2013,37(12),1903-1915, doi: 10.1111/ejn.12183.

The authors should clarify the medication used by the patients. How could the use of opioids have influenced their results (frontal lobe tests)? Could this explain, at least in part, that the worse group was that of chronic migraine patients (i. e, they consume more medication)?.

Thank you for the comments. We collected information on the number of times and days drugs were taken and the types of medication used, including analgesics, NSAIDs, ergot, triptans, and opioids. We enrolled 19 patients with chronic migraine with medication overuse headache who used the following medications: simple analgesics in seven patients, NSAIDs in five, triptans in two who also used NSAIDs, and a combination analgesics in seven. No patients taking opioids were enrolled in our study. We found significant correlations between days per month with headache and TMT variables or components of the WCST across the episodic and chronic migraine groups. However, other headache parameters (duration, intensity, days per month and number of times medication was taken) were not significantly correlated with frontal lobe-related neuropsychological variables (TMT, WCST, and IGT). Especially, only few subjects were enrolled in the medication overuse headache group in the study, so there is a limit to interpreting the results, and further research is needed. We have revised the following text in the Materials and Methods and Results as follows:

In addition, we recorded the number of days per month and the number of times medications were used as well as the types of medications used including analgesics, nonsteroidal anti-inflammatory drugs (NSAIDs), ergot, triptans, and opioids. (page 2, lines 95 – page 2, lines 98)

Of the 19 enrolled patients with CM+MOH, the types of medication used as follows; simple analgesics in seven patients, NSAIDs in five, triptans in two who also used NSAIDs, and a combination analgesics in seven. (page 4, lines 191 – page 4, lines 194)

Further, other headache parameters (duration, intensity, days per month and number of times medication was taken) were not significantly correlated with frontal lobe-related neuropsychological variables (TMT, WCST, IGT). (page 5, lines 224 – page 5, lines 226)

The test employed to assess the OF function, the IGT, does not correlate with any clinical or affective parameter, there are no differences between groups, and the standard deviations are important. Could this represent that this is a difficult to perform test and was performed at random by most of the participants?

We administered the computer-based IGT to all subjects to test decision-making mediated by orbitofrontal function. The IGT consists of picking cards from four decks, each of which can result in an unpredictable reward or penalty. Gains and losses occur in certain ratios. The sequence of gains and losses and the amount of each gain or loss encountered for each of the four decks are fixed. The IGT net scores range from -100 to +100. The standard deviation (SD) of the IGT score in our study was 21 for episodic migraineurs and 20 for chronic migraineurs, similar to previous results (SD range of ~16–25 for episodic or chronic migraineurs with medication overuse headache). We have added the following text to the Methods:

The sequence of gains and losses and the amount of each gain or loss encountered with each of the four decks are fixed [20].

(page 4, lines 159 – page 4, lines 160)

The IGT net scores range from –100 to +100.

(page 4, lines 166 – page 4, lines 167)

We have also added this citation:

  1. Bechara A. Iowa Gambling Task Professional Manual. Version 1, ed Raton B. Lutz: Psychological Assessment Resources, Inc. 2007.

Several paragraphs of the discussion do not add great information to this paper and can be dispensed with (lines 293-302).

As recommended, we omitted this paragraph in the Discussion.

We have tried to address the issues raised by the reviewers and editorial board member. We are grateful for the constructive comments that arose during the review process. We believe that our paper has been improved based on these suggestions.

Yours faithfully,
